# Host- and Age-Dependent Transcriptional Changes in *Mycobacterium tuberculosis* Cell Envelope Biosynthesis Genes after Exposure to Human Alveolar Lining Fluid

**DOI:** 10.3390/ijms23020983

**Published:** 2022-01-17

**Authors:** Anna Allué-Guardia, Andreu Garcia-Vilanova, Angélica M. Olmo-Fontánez, Jay Peters, Diego J. Maselli, Yufeng Wang, Joanne Turner, Larry S. Schlesinger, Jordi B. Torrelles

**Affiliations:** 1Population Health Program, Texas Biomedical Research Institute, San Antonio, TX 78227, USA; AgarciaVilanova@txbiomed.org (A.G.-V.); AOlmo@txbiomed.org (A.M.O.-F.); 2Integrated Biomedical Sciences Program, University of Texas Health Science Center at San Antonio, San Antonio, TX 78229, USA; 3Division of Pulmonary and Critical Care Medicine, School of Medicine, UT Health San Antonio, San Antonio, TX 78229, USA; PETERS@uthscsa.edu (J.P.); MaselliCacer@uthscsa.edu (D.J.M.); 4Department of Molecular Microbiology and Immunology, South Texas Center for Emerging Infectious Diseases, University of Texas at San Antonio, San Antonio, TX 78249, USA; Yufeng.Wang@utsa.edu; 5Host-Pathogen Interactions Program, Texas Biomedical Research Institute, San Antonio, TX 78227, USA; joanneturner@txbiomed.org (J.T.); LSchlesinger@txbiomed.org (L.S.S.)

**Keywords:** *Mycobacterium tuberculosis*, alveolar lining fluid (ALF), lung mucosa, cell envelope biosynthesis, gene expression

## Abstract

Tuberculosis (TB) infection, caused by the airborne pathogen *Mycobacterium tuberculosis* (*M.tb*), resulted in almost 1.4 million deaths in 2019, and the number of deaths is predicted to increase by 20% over the next 5 years due to the COVID-19 pandemic. Upon reaching the alveolar space, *M.tb* comes into close contact with the lung mucosa before and after its encounter with host alveolar compartment cells. Our previous studies show that homeostatic, innate soluble components of the alveolar lining fluid (ALF) can quickly alter the cell envelope surface of *M.tb* upon contact, defining subsequent *M.tb*–host cell interactions and infection outcomes in vitro and in vivo. We also demonstrated that ALF from 60+ year old elders (E-ALF) vs. healthy 18- to 45-year-old adults (A-ALF) is dysfunctional, with loss of homeostatic capacity and impaired innate soluble responses linked to high local oxidative stress. In this study, a targeted transcriptional assay shows that *M.tb* exposure to human ALF alters the expression of its cell envelope genes. Specifically, our results indicate that A-ALF-exposed *M.tb* upregulates cell envelope genes associated with lipid, carbohydrate, and amino acid metabolism, as well as genes associated with redox homeostasis and transcriptional regulators. Conversely, *M.tb* exposure to E-ALF shows a lesser transcriptional response, with most of the *M.tb* genes unchanged or downregulated. Overall, this study indicates that *M.tb* responds and adapts to the lung alveolar environment upon contact, and that the host ALF status, determined by factors such as age, might play an important role in determining infection outcome.

## 1. Introduction

*Mycobacterium tuberculosis* (*M.tb*), the causative agent of tuberculosis (TB), is one of the top leading causes of mortality worldwide due to a single infectious agent, with ~1.4 million attributed deaths in 2019 [1]. However, global estimates indicate that worldwide disruptions in the healthcare system during the current COVID-19 pandemic could lead to an additional 6.3 million new TB cases between 2020 and 2025 and an added 1.4 million more TB deaths [2,3]. Strict lockdowns have prevented patients from having access to TB medications and clinical evaluations, and have led to decreased TB diagnosis rates, since available resources have been redirected to prevent the spread of COVID-19 [4,5]. These factors are predicted to cause not only an increase in the number of TB cases, but also to promote the development of drug-resistant TB, stressing the need for the development of new anti-TB therapies [6].

Most current drugs target the *M.tb* cell envelope, a highly complex and dynamic structure comprised mainly of carbohydrates and lipids, which provide structural support and resistance to osmotic changes, as well as a critical immunoregulatory role during *M.tb* infection [7,8,9,10]. It consists of four main layers: (1) an inner plasma membrane with periplasmic space; (2) a peptidoglycan (PG) core covalently linked to arabinogalactan (AG) and mycolic acids (MAcs); (3) a peripheral layer of non-covalently linked lipids, glycolipids, and lipoglycans (e.g., phthiocerol dimycocerosates (PDIMs), trehalose dimycolate (TDM) and monomycolate (TMM), sulfolipids (SLs), phosphatidyl-*myo*-inositol mannosides (PIMs), lipomannan (LM), and mannose-capped lipoarabinomannan (ManLAM), among others), and; (4) the outermost layer or capsule [11,12]. The pathogenesis of *M.tb* is inherently linked to its heterogeneous and dynamic cell envelope surface, and cell envelope remodeling has been observed during infection and in response to environmental stresses [10]. Thus, the particular cell envelope composition of a mycobacterial cell at a given moment will define *M.tb*–host cell interactions and determine the infection outcome. However, it remains poorly understood how the *M.tb* cell envelope changes and adapts to the host-lung environment during the natural course of pulmonary infection, which is a critical gap in our knowledge for defining new drug targets against *M.tb* relevant to bacteria in the lung environment.

After the host inhales droplets containing *M.tb*, the airborne pathogen is deposited in the lung alveolar space. Here, it first comes in contact with soluble components of the alveolar lining fluid (ALF) for an undefined period of time (from minutes to hours/days) [13,14,15,16], before and after its encounter with host alveolar compartment cells such as alveolar macrophages (AMs) or alveolar epithelial cells (ATs) and immune cells such as neutrophils [17,18]. Our previous studies have demonstrated that homeostatic ALF hydrolytic enzymes, whose function is to promote lung health, can modify the *M.tb* cell envelope without reducing *M.tb* viability [19,20], including a reduction in major *M.tb* virulence factors mannose-capped lipoarabinomannan (ManLAM) and trehalose dimycolate (TDM) from the *M.tb* cell surface [19]. These ALF-derived *M.tb* cell envelope modifications have an impact on *M.tb* infection outcomes in vitro [19,20,21,22,23] and in vivo [22], since they allow for a better recognition by cells of the immune system and improved control of the infection. Indeed, exposure to ALF results in decreased *M.tb* association and intracellular growth within human macrophages, as well as altered intracellular trafficking and increased proinflammatory responses [19]. Neutrophils also possess an enhanced innate ability to recognize and kill intracellular ALF-exposed *M.tb*, while limiting excessive inflammatory responses [21]. In vivo infections using ALF-exposed *M.tb* demonstrate better control of infection in the mouse model [22]. Further, *M.tb* fragments released after ALF exposure by the action of the ALF hydrolases are capable of priming neutrophils [23] and modulating macrophages in an IL-10 dependent manner to contribute further to the control of *M.tb* [20].

Importantly, the levels and functionality of ALF soluble components are altered in certain human populations such as the elderly, with increased pro-oxidation and proinflammatory pathways, altered complement and surfactant levels, and decreased binding capability of surfactant protein A (SP-A) and D (SP-D) in the aging lung, defining what we call ‘dysfunctional ALF’ [24]. Consequently, *M.tb* exposed to elderly dysfunctional ALF, while maintaining its viability, show increased intracellular growth in macrophages and ATs [22,25], as well as increased bacterial burden in mice with increased lung tissue damage [22]. Altogether, these results indicate that the host ALF functional status plays a key role in shaping the *M.tb* cell envelope during the initial stages of infection. However, the overall impact of these different ALF microenvironments on *M.tb* adaptation to the human host and subsequent infection progression is still largely unknown.

Since hydrolytic enzymes present in functional human ALF (from healthy adult individuals) modify the *M.tb* cell envelope [19], we hypothesize that *M.tb* will compensate for these ALF-driven modifications by altering the expression of genes related to cell envelope biogenesis. Conversely, *M.tb* exposed to ALF with decreased functionality (such as elderly ALF) will show little to no changes. In addition, our published data [19,20] demonstrate that 15 min exposure to human ALF is enough to alter the *M.tb* cell envelope and its interactions with host cells, and that these modifications are maintained for up to 24 h [19,20]. In this study, we focus specifically on early *M.tb*–ALF interactions during the first stages of infection, and first aim to determine if short (15 min) or long (12 h) exposure to human ALF has any effects on the expression of targeted *M.tb* cell envelope genes, showing that gene expression is moderately altered at 15 min and that some transient expression changes happen at 12 h. Then, we use a multiplex qPCR assay to compare an extensive transcriptional profile of *M.tb* cell envelope genes associated with lipid, carbohydrate and amino acid metabolism, among others, after exposure to functional healthy adult (A) vs. dysfunctional healthy elderly (E) ALF. Our results show significant differences in gene expression, where A-ALF-exposed *M.tb* upregulated genes involved in cell envelope remodeling, thus implicating this remodeling in subsequent *M.tb*–host interactions during the infection process.

## 2. Results

### 2.1. Expression of Cell Envelope PIMs/LM/ManLAM Biosynthesis Genes in M.tb after Exposure to Human ALF

Our laboratory has previously demonstrated that ALF hydrolases modify the *M.tb* cell envelope, and that these cell envelope modifications occur in as little as 15 min of *M.tb* being in contact with ALF and are maintained up to 24 h, without compromising *M.tb* viability [19]. Importantly, these ALF-driven alterations of the *M.tb* cell envelope had an effect on *M.tb* infection outcomes in vitro [19,20,21,22,23] and in vivo [22]. Based on these results, we sought to determine if *M.tb* alters its cell wall biosynthetic pathways as a direct consequence of ALF exposure. One of the main *M.tb* cell envelope components, ManLAM, is decreased by the action of ALF hydrolases [19] and PIMs, LM and ManLAM are thought to be part of the same biosynthetic pathway. Thus, we first selected a few key *M.tb* cell envelope genes associated with biosynthesis of initial mannose donors GDP-Man (*manA*/*Rv3255c*, *manB*/*Rv3264c* (previously annotated as *manC*), *pmmA*/*Rv3257c* (previously annotated as *manB*), *pmmB*/*Rv3308*) and polyprenylphosphate-based mannose or PPM (*pmm1*/*Rv2051c*), the transcriptional regulator *whiB2/Rv3260c* [26], and mannosyl (*pimA*/*Rv2610c*, *pimB*/*Rv2188c*, *pimF*/*Rv1500*) and arabynosyl (*embC*/*Rv3793*) transferases (see Figure 1 and Table 1 in Materials and Methods) [27,28], all involved in this pathway. Expression was determined by RT-qPCR from *M.tb* with an intact and complete cell envelope that was exposed to individual human adult ALFs for 15 min and 12 h. Relative expression was normalized to housekeeping genes *rpoB* (Figure 2A) or *sigA* (Figure 2B). *M.tb* preparations exposed to the same heat-inactivated ALFs were used as reference samples.

After 15 min of exposure to different ALFs when compared to *M.tb* exposed to corresponding heat-inactivated ALFs (Figure 2), *pmmA* or phosphomannomutase A, previously annotated as *manB*, participates in the formation of Man-1-P from Man-6-P during biosynthesis of mannose donor GDP-Man (Figure 1), which is associated with the biosynthesis of mannosylated lypoglycans that bind to macrophage phagocytic receptors [32]. Although not statistically significant, *ppm1* (polyprenol-monophosphomannose synthase), which transfers mannose from GDP-mannose to endogenous PPM [33,34] in the PIMs/LM/ManLAM biosynthesis pathway, was moderately upregulated (fold change > 1), while *manA* and *embC* were slightly downregulated (fold change < 1, Figure 2). None of the targeted genes presented significant expression differences at 12 h post-ALF exposure compared to heat-inactivated ALF, although *ppm1* and *pimF* were slightly downregulated.

When comparing 15 min vs. 12 h post-ALF exposure, only *pmmA* decreased its expression significantly (Table 2). Other PIMs/LM/ManLAM biosynthesis-associated genes slightly decreased (*ppm1* and *pimF*), while others increased (*manA* and *embC*) (Table 2). Although most of the changes are not significant, the results suggest certain temporal and dynamic adaptation of *M.tb*’s cell envelope components in response to ALF, in agreement with previous reports showing cell envelope remodeling of *M.tb* during infection and in response to different environmental conditions [26,27,35,36,37,38,39,40]. *pmmB* expression was too low to be detected by our qPCR assay at either timepoints tested, thus no fold changes are shown.

### 2.2. Effects of A- vs. E-ALF Exposure on the Expression of Cell Envelope Biosynthesis Genes in M.tb

Since we observed some changes in the expression of key *M.tb* cell envelope genes in response to ALF and a dynamic response over time, and based on our previous publications showing that ALF status influences *M.tb*–host interactions after 12 h of exposure via multiple factors such as age [22,24,25], we next sought to determine if contact with different host ALFs would result in more pronounced changes and different transcriptional profiles of the *M.tb* cell envelope. We exposed *M.tb* Erdman with an intact cell envelope to individual A-ALFs or E-ALFs for 12 h, as described in our previous work [19], and calculated the relative expression of 83 genes related to cell envelope biosynthesis using a multiplex qPCR assay in a Biomark HD platform [41]. Results for some of the genes were compared to the previous targeted qPCR assay, showing no significant differences in gene expression between the two qPCR methods used in this study (Appendix A). Small differences observed could be attributed to inherent variability within human ALF, since we used different human donors for each of the assays.

Genes selected for the multiplex qPCR are related to the biosynthesis of key *M.tb* cell envelope lipid and carbohydrate components (Appendix A). For lipid metabolism, we targeted genes from the following pathways: fatty acid metabolism, glycerolipid and glycerophospholipid metabolism, phosphatidylinositol (PI, precursor for more complex glycolipids such as PIM and LAM), PIMs/LM/ManLAM biosynthesis, mycolic acid biosynthesis, and linoleic acid metabolism [9]. For carbohydrate metabolism, we included genes related to carbohydrate biosynthesis, glycolysis and gluconeogenesis, mannose and fructose metabolism, galactose metabolism, citrate cycle or TCA, glyoxylate and dicarboxylate metabolism, and inositol phosphate metabolism [42]. Gene names, with corresponding function, pathway and functional categories, as well as primer sequences, are listed in Appendix A. 

As shown in Figure 3A, most of the cell envelope genes associated with lipid metabolism from the different *M.tb*-targeted pathways were highly upregulated (fold change higher than 1.5) in *M.tb* exposed to A-ALFs (A_1_ to A_3_, from three different adult donors) when compared to the same *M.tb* strain exposed to E-ALFs (E_1_ to E_3_, from three different elderly donors), with A_1_-ALF having the highest changes in gene expression (Figure 3A and Appendix A). However, most of the changes did not reach statistical significance when comparing exposure of A-ALFs vs. E-ALFs, since A_1_-ALF-exposed *M.tb* presented much higher values when compared with *M.tb* exposed to the other two A-ALFs (attributed to the inherent human ALF variability) (Appendix A), increasing the standard deviation within the A-ALF group. Still, the expression trends are maintained and will be discussed here. Only a few of the lipid metabolism genes tested were downregulated (fold change below 0.67) in *M.tb* exposed to A-ALFs, including *fabH* and *fadD25* (FA metabolism, the latter involved in lipid degradation), and *adhE1*, *glpK*, and *cdsA* (glycerolipid and glycerophospholipid synthesis). ATP-binding cassette transporter Rv1747, thought to be involved in the export of lipooligosaccharides (LOS) through the mycobacterial membrane, was also downregulated in A-ALF-exposed *M.tb*, whereas negative regulator of Rv1747, named Rv2623 [43], showed increased expression (Figure 3A). Interestingly, *M.tb* exposed to two of the E-ALFs showed upregulation of Rv1747 (Figure 3A), contrary to A-ALF exposure, and although it did not reach statistical significance, it might indicate a potential increase in *M.t**b* PIM export after exposure to elderly ALF. All genes involved in mycolic acid synthesis tested in this study were upregulated in *M.tb* exposed to A-ALF. Finally, exposure to E-ALFs showed lesser effects on the overall expression of *M.tb* lipid metabolism when compared to A-ALFs, with several genes showing moderate upregulation only in E_3_-ALF-exposed *M.tb* (which presents higher expression values compared to the other E-ALF-exposed *M.tb* for most of the genes), and others genes with no effects or even decreased expression (Figure 3A). 

A similar trend was observed for carbohydrate metabolism genes (Figure 3B), where most of the genes from the different pathways studied showed increased expression in *M.tb* exposed to A-ALFs compared to E-ALFs (although only *pgmA* and *fucA* showed statistical significance, Appendix A). Particularly, 8 out of 12 genes associated with fructose metabolism and the mannose donor (GDP-Man/PPM) biosynthesis pathways were highly expressed in A-ALF-exposed *M.tb*, with only two genes downregulated (*pmmA*, which converts D-mannose 1-phosphate in D-mannose 6-phosphate, and *pfkA*, also a key enzyme involved in glycolysis) [44], and two other genes with variable expression across different A-ALF-exposed *M.tb* (*ppm1* and *fbA*) (Figure 3B). Galactose metabolism genes (*galK*, *galT*, *galU*, and *aglA*), essential for the biosynthesis of the cell envelope galactan core [45], were also upregulated after exposure to A-ALFs, while no changes or even decreased expression were observed in *M.tb* exposed to E-ALFs. Interestingly, predicted alpha-glucosidase *aglA* was increased after exposure to E_3_-ALF (Figure 3B). Further, most tricarboxylic acid cycle (TCA)-associated genes were upregulated by A-ALF exposure, with the exception of *sdhC* (membrane-anchored subunit of Sdh2, implicated in *M.tb* growth linked to the TCA cycle under hypoxia conditions) [46] which was downregulated upon both A- and E_1_-ALF *M.tb* contact. Glyoxylate and dicarboxylate genes showed decreased expression in E-ALFs when compared to A-ALF-exposed *M.tb*, whereas the only inositol phosphate gene tested *ino1*, catalyzing the first step in inositol synthesis for the production of major thiols and cell wall lipoglycans [47], had decreased expression after *M.tb* exposure to all ALFs tested, except for A_1_-ALF- and E_2_-ALF-exposed *M.tb*, where it was slightly increased (Figure 3B).

In addition to lipid and carbohydrate pathways, we studied genes belonging to other categories such as amino acid metabolism, redox homeostasis, and transcriptional regulators (Figure 3C and Appendix A). Results indicate that *M.tb* exposed to A-ALFs have major transcriptional changes compared with *M.tb* exposed to E-ALFs (although only *pyrC* and *eccB3* genes reached statistical significance when comparing the two ALF groups, Appendix A). Indeed, all genes associated with amino acid metabolism were highly upregulated in *M.tb* exposed to A-ALFs, except for *serC* (serine metabolism, Figure 3C) [48] which showed decreased expression. Other cell wall-associated genes were upregulated in *M.tb* exposed to A-ALFs including the putative membrane protein EccB3, part of the ESX-3 secretion system, important for zinc and iron uptake and homeostasis [49], and the cutinase precursor Cfp21, a lipolytic enzyme with immunogenic properties shown to elicit T- and B-cell responses [50,51]. Similarly, genes *ahpC* and *sodC*, involved in the oxidative stress response [52,53], and transcriptional regulators *dosR* and *phoP,* had increased expression in *M.tb* exposed to A-ALFs, whereas *virS* [54] did not show significant changes in expression. Overall, our results demonstrate that exposure to functional A-ALF results in broad changes in *M.tb* cell envelope biosynthesis, suggesting highly dynamic cell envelope changes with constant remodeling within the lung alveolar environment, whereas *M.tb* exposed to dysfunctional E-ALF showed more limited effects in gene expression.

## 3. Discussion

The cell envelope of *M.tb* is mainly composed of lipids and carbohydrates, and constitutes a dynamic structure known to adapt to the changing local host environment, especially during different stages of infection [8,10]. Human ALF contains hydrolases whose homeostatic function is to maintain lung health. We have demonstrated that healthy adult individuals have up to 17 hydrolase activities capable of altering the *M.tb* cell wall. Indeed, exposure of *M.tb* to these adult ALFs significantly alters the *M.tb* cell wall, reducing the content of two major cell envelope components, ManLAM and TDM, without compromising *M.tb* viability [19]. Importantly, these hydrolase activities are decreased in ALFs from healthy elders [24]. Thus, here, we demonstrate for the first time that exposure to human ALF, the first environment encountered by *M.tb* during early infection stages, results in transcriptional changes in key *M.tb* cell envelope biogenesis genes. Indeed, exposure to healthy A-ALFs resulted in increased expression of most of the *M.tb* cell envelope-associated genes from lipid, carbohydrate and amino acid metabolic pathways, among others. In contrast, *M.tb* exposed to E-ALFs from healthy elderly donors, which we demonstrated constitutes a more oxidized, proinflammatory and dysfunctional environment [22,24], did not show many effects on gene expression. 

We first assessed essential genes from the PIMs/LM/ManLAM synthesis and the mannose metabolism pathways (carbohydrate metabolism) (Figure 1), since our previous studies showed a decrease in ManLAM in the cell envelope of *M.tb* after ALF exposure [19]. PIM/LM/ManLAM molecules are essential for regulating *M.tb* recognition, uptake, survival and modulating the host immune response [55,56]. Indeed, ManLAM has been shown to block phagosome–lysosome (P–L) fusion by inhibiting the Ca^2+^/Calmodulin phosphatidyl-inositol-3-kinase (PI3K) hvps34 pathway, promoting *M.tb* intracellular survival [57,58,59]. The biosynthesis of these molecules depends on mannose donors such as GPD-Man and PPM. Previous studies suggest that mannose donor levels are altered during the course of the infection. Indeed, *M.tb* mannose donor biosynthesis genes had increased expression levels 2 h after macrophage infection, and then gradually decreased [26]. These were also found upregulated in an in vitro granuloma model [27]. In our study, most of the genes involved in the PIM/LM/ManLAM biosynthesis pathway were upregulated after *M.tb* Erdman was exposed to A-ALFs (Figure 3A,B). We speculate that *M.tb* is trying to compensate for the loss of ManLAM and other mannose-containing cell envelope surface components due to the action of A-ALF hydrolases [19], with major implications for disease progression. In this regard, bacilli exposed to functional A-ALF might be taken up by antigen-presenting cells (APCs) before they can reconstitute ManLAM and be cleared, while bacilli that rapidly upregulate and replenish ManLAM on the cell surface before its encounter with APCs will be able to block P–L fusion and survive within the host cells. Conversely, exposure to E-ALFs did not have major transcriptional effects in genes from both mannose donors and PIM/LM/ManLAM biosynthesis pathways (Figure 3A,B), likely because E-ALF have fewer hydrolase activities [24]. This E-ALF deficiency in hydrolase activities could be directly linked to E-ALF being a highly oxidative environment [22,24], therefore impacting the *M.tb* cell envelope and subsequent remodeling to a much lesser degree during infection. Since removal of surface lipids in *M.tb* enhances trafficking to acidic compartments [58], fewer ALF-driven alterations after E-ALF exposure might partially explain why *M.tb* replicates faster in the elderly lung environment by residing in its protective intracellular niche [22,25]. Further studies will be necessary to determine the specific impact of elderly ALF hydrolases on the *M.tb* cell envelope. 

Most lipid metabolism genes associated with mycolic acid synthesis, fatty acid metabolism, glycerolipid and glycerophospholipid synthesis, and linoleic acid metabolism, were also highly upregulated in A-ALF- compared to E-ALF-exposed *M.tb* (Figure 3A). A similar trend was observed for carbohydrate metabolism, amino acid metabolism, and genes involved in redox homeostasis, stress response, and transcriptional regulation (Figure 3B,C). Nitrogen and amino acid metabolism are important for *M.tb* pathogenesis and host colonization during infection, where intracellular bacteria exploit host nitrogen sources for growth and replication [60,61]. Importantly, amino acids acquired from the host, such as Ala and Gly, might be directly assimilated for the synthesis of cell wall biomass and incorporated into the PG layer [62,63,64]. Transcriptional regulators DosR and PhoP also showed increased expression upon contact with functional A-ALF. These proteins are part of the two-component system implicated in a large number of *M.tb* adaptive responses [65,66], playing key roles as regulators of *M.tb* virulence [67,68]. Interestingly, DosR has been shown to play a role in lipid accumulation during oxidative stress and iron starvation in certain *M.tb* clinical strains [69,70]. 

These results indicate that the *M.tb* response to the ALF environment is not only limited to *M.tb* cell envelope remodeling, but also potentially affects overall *M.tb* metabolism and virulence [19,20,21,22,23,24,25,71], with major implications in infection progression and TB disease outcome. Since this study is only focused on the expression of cell envelope genes, ongoing global transcriptomic studies using RNA-seq technologies will provide clues as to which other metabolic pathways are altered by the human ALF environment, with the potential to decipher key upregulated bacterial determinants during early stages of infection that can be targeted for the development of new preventative and therapeutic strategies [72]. Indeed, *M.tb* is exposed to ALF during the first infection stages upon deposition to the alveolar space, which is studied here, but also when escaping from necrotic cells or in cavities during active TB episodes leading to transmission [17]. Future studies will focus on the role of the ALF in the adaptation of *M.tb* to the granuloma environment and the reactivation process. Thus, it is plausible that *M.tb* adapts its cell envelope to the alveolar environment, upregulating the expression of specific genes to compensate the changes generated upon contact with ALF hydrolases, and determining subsequent interactions with host cells. In this regard, timing is expected to be important, as *M.tb* bacilli not able to restore its cell wall constitution before encountering antigen-presenting cells, such as alveolar macrophages, might be cleared [19,20,21].

Key genes such as Rv1747, thought to participate in the export of PIMs to the cell envelope through negative regulation by stress protein Rv2623 [43], were highly downregulated in *M.tb* exposed to A-ALFs (fold changes of 0.11, 0.14, and 0.09 for A_1_, A_2_, and A_3_, respectively). Indeed, an ΔRv2623 mutant showed enhanced PIM expression and was hypervirulent in mice [73], while ΔRv1747 had decreased levels of PIMs and was growth-attenuated [74]. Our data suggest that contact with functional A-ALF reduces the expression of PIM transporter Rv1747 through increased expression of Rv2623, modulating the export of immunomodulatory PIMs and potentially influencing bacterial growth and virulence. 

Taken together, these results suggest that *M.tb* compensates for the loss of cell surface components (due to the action of ALF hydrolases) [19] by upregulating and activating different cell envelope biosynthesis pathways to rebuild its cell wall, at the detriment of downregulating some key genes (e.g., Rv1747) involved in the transport of cell envelope components to the surface [43]. This shift between ALF innate homeostatic mechanisms and *M.tb* countermeasures in the ALF microenvironment, dependent on the host ALF status (A-AF vs. E-ALF), will likely determine subsequent interactions between *M.tb* and host cells, as well as intracellular trafficking and infection outcomes [19,20,21]. Further studies are needed to provide better insight to why E-ALF-exposed *M.tb* replicates faster than A-ALF-exposed *M.tb* in both professional and nonprofessional phagocytes, and to explain why E-ALF status in old age enhances *M.tb* infection in vitro [22,24,25] and in vivo [22] and contributes to elders being more susceptible to respiratory infections in general. Finally, it is important to consider that the cell envelope composition of *M.tb* is strain-specific, with differences observed in *M.tb* strains from different lineages, and thus, different ALF-driven alterations in *M.tb* metabolism may drive different infection progression [75]. In addition, it is possible that the cell envelope of cultured *M.tb* might differ from bacteria transmitted via aerosol from active TB patients, and that other potential ALF-driven alterations might not have been captured in this study, which requires further investigation. 

In summary, our study provides evidence that *M.tb* contact within the ALF shapes the composition of its cell envelope, which, depending on the ALF status (‘functional A-ALF’ vs. ‘dysfunctional E-ALF’) [22,24,25], is likely to define subsequent *M.tb*–host cell interactions. Indeed, E-ALF-exposed *M.tb* presented minimal transcriptional changes when compared to A-ALF-exposed *M.tb*, which we speculate provides a fitness advantage to *M.tb* as its cell wall remains intact, and thus, it has the energy reserves required to efficiently infect and replicate faster within host cells of elderly individuals. In contrast, *M.tb* undergoes significant alterations on its cell wall (significant loss of virulent factors ManLAM and TDM, among others) upon exposure to A-ALF [19]. This triggers a greater transcriptional change that we interpret as efforts of A-ALF-exposed *M.tb* to reprogram its metabolism to quickly rebuild its cell wall, specifically upregulating genes involved in the biosynthesis of *M.tb* virulence factors such as ManLAM, with the ensuing energy requirements. This can be detrimental for A-ALF-exposed *M.tb* and favor host cells to control infection better in adult individuals. Future studies will investigate the metabolic status of E-ALF-exposed *M.tb* after infection of professional and nonprofessional phagocytes in vitro and in vivo, and will correlate bacterial and host determinants associated with increased susceptibility to infection in old age.

## 4. Materials and Methods

### 4.1. Human Subjects and Ethics Statement

Human ALFs used in this study were previously isolated from collected bronchoalveolar lavage fluid (BALF) from healthy adult and elderly volunteers, in strict accordance with the US Code of Federal and approved Local Regulations (The Ohio State University Human Subjects IRB numbers 2012H0135 & 2008H0119 and Texas Biomedical Research Institute/UT-Health San Antonio/South Texas Veterans Health Care System Human Subjects IRB numbers HSC20170667H & HSC20170673H), and Good Clinical Practice as approved by the National Institutes of Health (NIAID/DMID branch), with written informed consent from all human subjects. All healthy adults (18–45 years old) and elderly (+60 years) were TST- or IGRA-negative, and were recruited from both sexes (50:50 male:female ratio) with no discrimination based on race or ethnicity (including Hispanic, American Indian/Alaska native, Asian, Native Hawaiian/Other Pacific islanders, Black or African American, and White). Donors with the following comorbidities were excluded: smokers, drug and alcohol users, asthma, acute pneumonia, upper/lower respiratory tract infections, acute illness/chronic condition, heart disease, diabetes, obesity, obstructive pulmonary disease (COPD), renal failure, liver failure, hepatitis, thyroid disease, rheumatoid arthritis, immunosuppression or taking nonsteroidal anti-inflammatory agents, human immunodeficiency virus (HIV)/AIDS, cancer requiring chemotherapy, leukemia/lymphoma, seizure history, blood disorders, depression, lidocaine allergies (used during the BAL procedure), pregnancy, nontuberculous mycobacterial infection, and TB.

### 4.2. Collection of BALF and ALF

BALF was collected from healthy adult or elderly donors in sterile endotoxin-free 0.9% of NaCl, using a dwell/collection time of 5 s for each lavage with a large volume (50 mL). Collection from the lower airways ensures that most of the lavage is of alveolar origin (ALF). BALF was then filtered through 0.2 μm filters, and further concentrated 20-fold using Amicon Ultra Centrifugal Filter Units with a 10-kDa molecular mass cut-off membrane (Millipore Sigma, Burlington, MA, USA) at 4 °C to obtain ALF with a physiological concentration reported within the human lung (1 mg/mL of phospholipid), as we previously described [19,20,21,22,23,24,71]. ALF was aliquoted in low protein-binding tubes and stored at −80 °C until further use. ALFs used in this study were selected based on a 50:50 male:female ratio and no discrimination of race or ethnicity. ALF composition and function from adult and elderly individuals has been previously published [19,20,21,22,23,24], and thus, not determined in this study. 

### 4.3. Bacterial Cultures and ALF Exposure 

*M.tb* strains GFP-Erdman (kindly provided by Dr. Marcus Horwitz, UCLA) and H_37_R_v_ (ATCC# 25618) were cultured in 7H11 agar plates (BD BBL), supplemented with oleic acid, albumin, dextrose and catalase (OADC) at 37 °C for 14 days. Single bacterial suspensions (~1 × 10^9^ bacteria/mL) were obtained as previously described [19,20,21,22,23]. Bacterial pellets with an intact and complete cell envelope were exposed to individual ALFs (from different donors) for 15 min or ~12 h at 37 °C. Although it is currently unknown how long *M.tb* stays in contact with ALF prior to encountering host cells (each infection process might be different depending on multiple bacterial and host factors), our previous publications [19] show that the effect of ALF hydrolases on the *M.tb* cell envelope and subsequent interactions with host cells are independent of their action time (from 15 min to 12 h) [19], and we have consistently used 12 h of ALF exposure in all our published works [19,20,21,22,23,24,25]. After exposure, ALF was removed and bacterial pellets were directly incubated in RNAProtect (Qiagen, Hilden, Germany) for 10 min at room temperature (RT), centrifuged at 13,000× *g* and stored at −80 °C until further use. For each of the ALFs, corresponding heat-inactivated ALFs (2 h at 80 °C) [21] were used as controls in parallel.

### 4.4. RNA Extraction and cDNA Synthesis

RNA from ALF-exposed bacterial pellets was extracted using the Quick-RNA Fungal/Bacterial Miniprep kit (Zymo Research, Irvine, CA, USA), following the manufacturer’s protocol. Briefly, bacterial pellets were resuspended in lysis buffer and transferred to a ZR BashingBead Lysis tube containing 0.1 mm and 0.5 mm ceramic beads. The bead-beating procedure was performed to break the tough-to-lyse mycobacterial cell envelope in a Disruptor Genie (10 cycles of 1 min at maximum speed with 1 min intervals on ice). RNA was isolated from the supernatant using Zymo-spin columns, including an in-column DNAse I treatment, and eluted in nuclease-free water. To completely remove the genomic DNA, a second DNAse treatment was performed on the isolated RNA using TURBO DNAse reagent (Thermo Fisher Scientific, Waltham, MA, USA) for 30 min at 37 °C. Final RNA concentration and quality were measured with a Qubit 4 Fluorometer using the HS RNA kit and a Nanodrop One^C^ (Thermo Fisher Scientific, Waltham, MA, USA), respectively. RNA (500 ng) was used for the synthesis of cDNA using the RevertAid H Minus First Strand cDNA Synthesis kit (Thermo Fisher Scientific, Waltham, MA, USA) with random hexamer primers, following the manufacturer’s guidelines.

### 4.5. qPCR Analysis of Targeted Genes

Real-time quantitative PCR (qPCR) was performed to measure the expression of 10 genes associated with the *M.tb* cell wall biosynthesis pathways (*pmmA*, *manB*, *whiB2*, *pimA*, *pimB*, *pimF*, *embC*, *pmmB*, *pmm1*, and *manA*) after exposure of *M.tb* H_37_R_v_ to individual healthy human ALFs (*n* = 6 biological replicates, from different donors). cDNA and primers were used in a 20 μL qPCR reaction with PowerUp SYBR Green Master Mix (Applied Biosystems, Waltham, MA, USA), following manufacturer’s instructions. Reactions were run in an Applied Biosystems 7500 Real-Time PCR instrument with the following settings: reporter SYBR Green, no quencher, passive reference dye ROX, standard ramp speed, and continuous melt curve ramp increment. Expression was calculated relative to housekeeping genes *rpoB* or *sigA* using the 2^−ΔΔCT^ method [76]. 

### 4.6. High-Throughput Multiplex qPCR

Primer pairs for multiplex qPCR assay were designed using BatchPrimer3 (https://wheat.pw.usda.gov/demos/BatchPrimer3/) [77] (accessed on 16 July 2019) and PrimerQuest (IDT, https://www.idtdna.com/pages/tools/primerquest) (accessed on 18 July 2019), with the following settings: primer size 18–21 ntds, T_m_ of ~54–60 °C, and maximum 3′ self-complementarity of 3 ntd. The best primers within these parameters were selected and aligned to the *M.tb* H_37_R_v_ reference genome (Genbank accession number: NC_000962.3) [78,79,80] to confirm that they uniquely aligned to the targeted gene regions. A high-throughput multiplex qPCR targeting more than 80 genes associated with *M.tb* metabolism and cell wall biogenesis was performed on *M.tb* Erdman exposed to individual ALF using the Biomark 96.96 Dynamic Array IFC for Gene Expression in a Biomark HD instrument (Fluidigm, South San Francisco, CA, USA). Briefly, 1.25 μL of cDNA was preamplified in a 5 μL reaction with our pool of specific primers (500 nM) using the Fluidigm Preamp Master Mix for a total of 10 cycles. Then, a 1/10 dilution of the preamplified cDNA and 100 μM of combined forward and reverse primers were used to prepare the sample premix and assay mix, respectively. The 96.96 IFC chip was loaded and the assay was run in a Biomark HD following the manufacturer’s instructions for Gene expression using Delta Gene Assays (Fluidigm, South San Francisco, CA, USA). Relative expression for each of the genes was calculated in the Fluidigm Real-Time PCR Analysis software v4.5.2 (Fluidigm, South San Francisco, CA, USA) using the 2^−ΔΔCT^ method with *rpoB* as the reference gene, and reported as fold changes of individual A-ALF (*n* = 3 biological replicates using ALFs from different donors) or E-ALF (*n* = 3 biological replicates using ALFs from different donors) exposed *M.tb* Erdman compared to control samples (*M.tb* Erdman exposed to corresponding heat-inactivated ALFs). As a control to compare both methods (Biomark vs. targeted qPCR), we included *M.tb* H_37_R_v_ exposed to adult ALF in the Biomark multiplex qPCR assay.

### 4.7. Statistical Analysis

Statistical significance between the two qPCR methods used in this study was calculated in GraphPad Prism v9.0.1 (GraphPad Software, San Diego, CA USA, www.graphpad.com) for each of the genes, with a two-way ANOVA using the Sidak’s correction for multiple comparisons test with a 95% confidence interval. Statistical significance of ALF-exposed *M.tb* vs. heat-inactivated ALF-exposed *M.tb* (targeted qPCR assay) was calculated using the ΔCt values with a two-way ANOVA for multiple comparisons with an uncorrected Fisher’s LSD test. The significance of A-ALF-exposed *M.tb* vs. E-ALF-exposed *M.tb* (high-throughput multiplex qPCR) was also calculated with a two-way ANOVA for multiple comparisons with an uncorrected Fisher’s LSD test. * *p*-value < 0.05; ** *p*-value < 0.005; *** *p*-value < 0.0005; **** *p*-value < 0.00005.

## Figures and Tables

**Figure 1 ijms-23-00983-f001:**
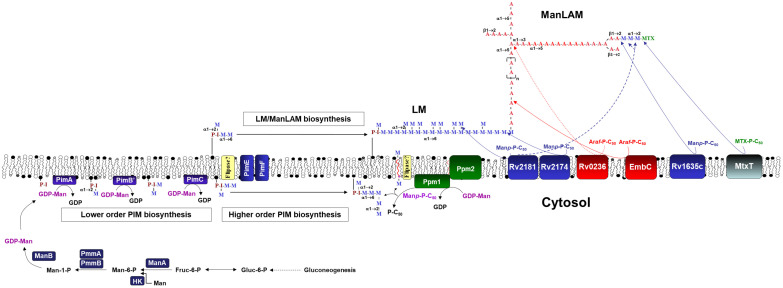
*M.tb* GDP-Man, PPM and PIM/LM/ManLAM biosynthetic pathways. GDP-Man can be biosynthesized directly from gluconeogenesis, through formation of Fruc-6-P from Glc-6-P (by the action of Glc-6-P isomerase), formation of Man-6-P (by the action of Man-6-P isomerase or ManA/Rv3255c), formation of Man-1-P (by the action of phosphomutases PmmA/Rv3257c, previously annotated as ManB, and PmmB/Rv3308), and finally formation of GDP-Man (by the action of Man-1-P guanylyn-transferase or ManB/Rv3264c, previously annotated as ManC). Further, Man-6-P can also be directly formed from Man by the action of a hexokinase (HK). PPM is formed from GDP-Man by the action of polyprenyl monophosphomannose synthase or Ppm1/Ppm2). Further, CDP-DAG together with inositol by the action of PI synthase (Rv2612c) forms PI, which is further mannosylated by several mannosyltransferases (PimA to PimF) to generate higher order PIMs using GDP-Man and PPM as mannose donors. At one point, from PIM_4_ and using PPM as the major mannose donor, PIM_4_ is heavily mannosylated by an undisclosed number of mannosyl transferases generating LM, and further arabinosylated with arabinosyl transferases generating LAM, which can be further mannose-capped by the mannosyl transferases action. LAM can also contain methylthio-_D_-xylose (MTX) capping motifs [29,30], where MtxT (Rv0541c) transfers MTX to the mannoside caps of LAM [31]. Note: For simplicity, acyltransferases (e.g., Rv2610c) are not depicted, and neither is the formation of MTX-P-C50 by MtxS (Rv0539).

**Figure 2 ijms-23-00983-f002:**
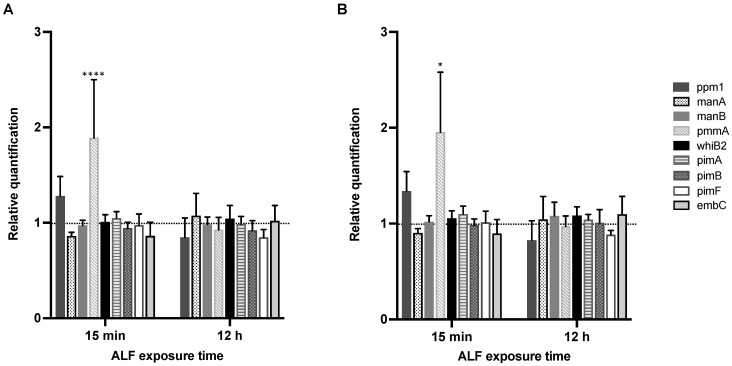
Relative expression of selected PIM/LM/ManLAM biosynthesis genes in *M.tb* H_37_R_v_ after exposure to ALF. *M.tb* was exposed for 15 min and 12 h to individual healthy human ALFs (*n* = 6 biological replicates, from six independent donors), using *rpoB* (**A**) or *sigA* (**B**) as reference genes. Expression values are shown as fold changes, and were calculated using the 2^−ΔΔCT^ method (ALF-exposed *M.tb* vs. corresponding heat-inactivated ALF-exposed *M.tb*) and plotted as the mean ± SEM using Prism v9. Statistical significance between ALF-exposed *M.tb* and heat-inactivated ALF-exposed *M.tb* is shown for each of the genes and timepoints for both housekeeping genes; ns: not significant; * *p*-value < 0.05; **** *p*-value < 0.00005.

**Figure 3 ijms-23-00983-f003:**
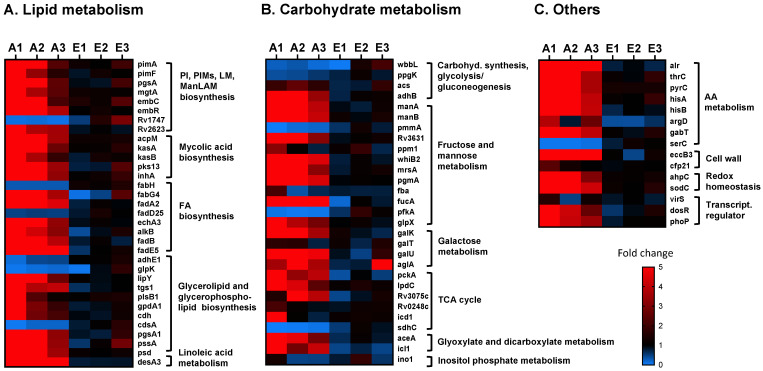
Relative expression of cell envelope biogenesis and metabolism genes in *M.tb* Erdman exposed to A-ALF and E-ALF. Heatmap showing relative expression of cell wall genes associated with (**A**) lipid metabolism; (**B**) carbohydrate metabolism; and (**C**) other pathways in *M.tb* after being exposed to individual A-ALFs (*n* = 3 biological replicates, A_1_- to A_3_-ALFs), or to E-ALFs (*n* = 3 biological replicates, E_1_- to E_3_-ALFs). Expression was normalized using *rpoB* as reference gene and calculated using the 2^−ΔΔCT^ method (ALF-exposed *M.tb* vs. heat-inactivated ALF-exposed *M.tb*). Heatmap was constructed using Prism v9, with downregulated genes in blue (0-to-1-fold changes) and upregulated genes in red (1-to-5-or-more-fold changes). Genes are grouped based on their assigned pathways (see Appendix A). Note that genes *pimB*, *accD3*, *adhC* (lipid metabolism), *pmmB*, Rv0794c (carbohydrate metabolism), and *metZ* (others) were below the limit of detection in one or more of the samples, and have not been included in the heatmaps.

**Table 1 ijms-23-00983-t001:** List of primers targeting the PIMs/LM/ManLAM biosynthesis pathways in *M.tb*.

Locus Tag H_37_R_v_	Gene Name	Product and Function	Primer Sequence (5′ to 3′)	Tm (°C)	Ref.
Rv0667	*rpoB*	Reference gene	rpoB-F: CCTGGAAGAGGTGCTCTACG rpoB-R: GGGAAGTCACCCATGAACAC	60 60	[26] [26]
Rv2703	*sigA*	Reference gene	sigA-F: CTCGGTTCGCGCCTACCTCA sigA-R: GCGCTCGCTAAGCTCGGTCA	68 68	[28] [28]
Rv3255c	*manA*	mannose-6-phosphate isomerase/GDP-Man biosynthesis: formation of Man-6-P	manA-F: GTTCACCACCTGGATTACCG manA-R: AACCCTCGGTGCATAACAAG	60 60	[26] [27]
Rv3264c	*manB* ^a^	D-alpha-D-mannose-1-phosphate guanylyltransferase/GDP-Man biosynthesis: formation of GDP-Man	manB-F: ACATCGCCGTTAAACACCAT manB-R: GTTCCTCACCCATCTGCTGT	60 60	[27] [27]
Rv3257c	*pmmA* ^b^	phosphomannose mutase/GDP-Man biosynthesis: formation of Man-1-P	pmmA-F: GATCACGTTGTGGATGATGG pmmA-R: GTGGATCTGCAGGCCTATGT	60 60	[26] [27]
Rv3308	*pmmB*	phosphomannose mutase/GDP-Man biosynthesis: formation of Man-1-P	pmmB-F: ATACAGATCACGGCGTCACA pmmB-R: CGCTGGATATAACGGTCGAT	60 60	[27] [27]
Rv2051c	*ppm1*	Polyprenol-monophosphomannose synthase/PPM biosynthesis	pmm1-F: TGGTTGAAGTCGATCCTTCC pmm1-R: GCGAACAAGACCAGGCATATG	60 63	[26] [26]
Rv3260c	*whiB2*	Transcript. regulatory protein	whiB2-F: CCATTCGAGGAACCTCTGC whiB2-R: CAGGGCGTACTCCAGACACT	61 60	[26] [26]
Rv2610c	*pimA*	alpha-(1-2)-phosphatidylinositol mannosyl-transferase/PIM biosynthesis (1st step)	pimA-F: CCGCACTGCCTGATTACTTT pimA-R: CGGCTCGTGTAGATGCAGTA	60 60	[27] [27]
Rv2188c	*pimB*	alpha-(1-6)-phosphatidylinositol mannosyl- transferase/PIM biosynthesis (2nd step)	pimB-F: CTCGGTGGTCAAGGTACTCG pimB-R: GTGGTCACCTTTGGGAACAT	61 60	[27] [27]
Rv1500	*pimF*	glycosyltransferase/LM/ ManLAM biosynthesis	pimF-F: CGCCGACGTAGTATTTGGTT pimF-R: TGCGTACATAGTCGGCTGTC	60 60	[27] [27]
Rv3793	*embC*	Arabynosyl-tranferase/ManLAM biosynthesis	embC-F: ATCACCGAGCTGCTGATG embC-R: TGCGAGTCACCGTTCCTA	58 59	[28] [28]

^a^ Previously annotated as *manC*. ^b^ Previously annotated as *manB*.

**Table 2 ijms-23-00983-t002:** Comparison of relative expression between 15 min and 12 h of ALF exposure for targeted *M.tb* genes.

Genes	15 min vs. 12 h (*rpoB*)	15 min vs. 12 h (*sigA*)	Statistical Significance
*ppm1*	−0.4330	−0.5129	ns
*manA*	0.2125	0.142	ns
*manB*	0.01161	0.0622	ns
*pmmA*	−0.9634	−0.9820	***
*whiB2*	0.03380	0.0271	ns
*pimA*	−0.06219	−0.0576	ns
*pimB*	−0.02300	0.0218	ns
*pimF*	−0.1283	−0.129	ns
*embC*	0.1572	0.2012	ns

Differences in the expression of targeted *M.tb* cell envelope genes between 15 min and 12 h after ALF exposure were calculated (fold change at 12 h–fold change at 15 min) for each reference gene. Statistical significance between 15 min and 12 h for each of the genes was calculated with a two-way ANOVA for multiple comparisons with an uncorrected Fisher’s LSD test. *** *p*-value < 0.0005; ns: nonsignificant.

## Data Availability

Not applicable.

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
