# Peer review of "Host- and Age-Dependent Transcriptional Changes in *Mycobacterium tuberculosis* Cell Envelope Biosynthesis Genes after Exposure to Human Alveolar Lining Fluid"

_ijms, 2022, doi:10.3390/ijms23020983_

Round 1
Reviewer 1 Report
The submitted manuscript presents the results of gene transcription analysis of M.tuberculosis exposed to alveolar lining fluid from healthy subjects and from elder people. The study proves the changes in the bacteria metabolism which are induced by the treatment and lead to recompensation mechanisms.
The undertaken study has solid basis justified by previous studies. The working hypothesis is correct and it is verified within this manuscript. Experiments are correct. The presentation of results is detailed, supported by tables and figures. Discussion is current and extensive.
The main aim of the study was to observe changes at the transcriptomic level, therefore the work is suitable for the IJMS and additional experiments monitoring metabolite changes would be of value, however are not mandatory at this stage.
Some questions asked during the first submission could be however answered to clarify the authors poin of view, and this will be with benefit for readers.
I suggest to address points risen in first submission (not to perform additional experiments).
Author Response
We thank the reviewer for the time taken to review our manuscript and for his/her positive comments. Below are the answers to the questions raised by the reviewers during the first submission, and how these have been addressed in the revised version of the manuscript, point by point:
Specific comments:
1. What does the cell envelope mean in this study exactly? Does the concept include capsule or only so called mycomembrane?
Response: This study includes the complete M.tb cell envelope (including the outer material or capsule). This has been clarified in the manuscript (lines 141-142, line 187, line 460). The different layers and components of the complete M.tb cell envelope are described in lines 53-59.
2. The authors describe that ALFs were collected from BALF. However, BALF may contain not only alveolar lining fluid but also epithelial lining fluid. How did the authors prevent the contamination of fluid other than ALF? Did the authors confirm the biochemical components of the BALF specimen?
Response: Although we cannot ensure that BAL fluid contains 100% ALF, during BAL procedure we use a dwell/collection time of 5 seconds for each lavage with a large volume (50 ml), collecting from the lower airways, which ensures that most of the lavage is of alveolar origin (ALF). We have used the same procedure in all our previous publications. The ALF nomenclature was used after a careful discussion involving many pulmonologists members of the American Thoracic Society (ATS). This has been clarified in the methods section (lines 444-446).
We have not included the biochemical description of the ALFs used here, since we have previously published the ALF composition and function in human ALF from elderly individuals compared to adult individuals, as well as the effects of its exposure in vitro (in human monocyte-derived macrophages or MDMs and alveolar epithelial cells) and in vivo (mouse model) (Moliva et al., 2014; Moliva et al., 2019, Olmo-Fontanez et al., 2021; Garcia-Vilanova et al., 2021). The aim of this current study is to focus on the M.tb cell envelope transcriptional changes rather than the host, following up on our previous publications describing ALF differences between adult and elderly individuals, and the observed changes in the cell envelope surface of M.tb (Arcos et al., 2011). This has been clarified in lines 452-454.
3. Are the mycomembrane structures of cultured M. tuberculosis and those in generated bioaerosol from active tuberculosis patient same? Is the use of cultured M. tuberculosis in this study relevant?
Response: The reviewer raised a valid point. We agree that the cell envelope of cultured M.tb might differ from the cell envelope of transmitted bacteria via aerosol. Although we do not know the effects of ALF on M.tb’s transmissibility in humans, we have previously showed that young mice infected via aerosol with E-ALF-exposed M.tb present an exacerbated M.tb infection in the lungs, demonstrating that exposure to E-ALF definitely impacts the infection outcome (Moliva et al., 2019). Since we cannot obtain ALF and/or bacteria from active TB patients (and isolating enough bacteria from active TB patients would still require additional culturing steps), our best approach is to perform ALF exposure from cultured bacteria. We added a comment in the discussion to highlight this limitation (lines 394-397).
4. Figure 1 and Table 1 should move to materials and methods section.
Response: We have now moved Table 1 to the Materials and Methods section. We kept Figure 1 in the Results section, so that expression results for the PIMs/LM/ManLAM pathway genes can be easily interpreted with the visual aid of Figure 1, which contains the position and function of each gene in the LAM biosynthetic pathway.
5. It is not clear from this manuscript whether the authors mixed all ALF specimens and made up pooled stock or used individual ALF. It is surely important to make this point clear. If latter, how did the authors choose the ALFs use in this study?
Response: We exposed M.tb to individual ALFs. This has been now clarified throughout the manuscript. The selection of ALFs was based on a 50:50 male:female ratio and different donor ethnicities within each age range (selection criteria has now been added to the Materials and Methods, lines 451-452).
6. There is no demographic or personal data of individuals from whom the ALF samples were collected. Did the authors perform IGRA to all participants; no IGRA positives?
Response: An IGRA test was performed on all participants, and all ALFs used in this study were from IGRA negative individuals. We have now included demographic information and donor exclusion criteria in the Materials and Methods section (lines 430-441).
7. Why did the authors perform targeted gene expression analysis? RNAseq will provide more deep insights about the influence of ALF to mycobacterial components, as the authors described in discussion section.
Response: We agree with the reviewer that RNA-seq will definitely provide better insights of the complete transcriptome profile of M.tbafter ALF exposure. Since previous work in the lab showed differences in the M.tb cell envelope surface after human ALF exposure (Arcos et al., 2011), for this particular study we decided to focus only on the cell envelope and see if these surface changes would also translate to altered expression of the cell envelope biosynthesis pathways by using a targeted approach. Since the resulting data from the study suggests that the M.tb response to ALF is not only limited to cell envelope remodeling but other metabolic pathways, we are currently investigating the complete M.tb transcriptional profile after ALF exposure to determine which other genes/pathways might be impacted, as well as how ALF-exposed bacteria adapts to the host environment during infection. This is an ongoing study and will be submitted for peer-review in a follow-up manuscript. A comment regarding future RNA-seq studies has been added to the discussion (lines 351-356).
8. 15 min exposure seems reasonable, but 12 h seems too long. No cellular immunity response happens within 12 h in human alveolar region?
Response: The reviewer raised a valid point. It is currently unknown how long M.tb stays in contact with ALF prior to encountering host cells, since each infection process might be different and depending on multiple bacterial and host factors. However, previous publications from our lab (Arcos et al., 2011) show that the effect of ALF hydrolases on the M.tb cell envelope and subsequent interactions with host cells are independent of their action time (from 15 min to 12 h) (Arcos et al., 2011), and we have consistently used 12 h of ALF exposure in all our published studies (Arcos et al., 2017; Arcos et al., 2015; Scordo et al., 2017; Scordo et al., 2019, Moliva et al., 2018; Moliva et al., 2019; Olmo-Fontanez et al., 2021). Additionally, each alveolar sac is estimated to have 8-12 alveolar macrophages, 40 alveolar epithelial cells type I and 67 alveolar epithelial cells type II (Stone et al., 1992 PMID: 1489139; Torrelles and Schlesinger, 2017 PMID: 28366292), with and ALF recycling time of ~ 18h. Thus, it is plausible that when M.tb reaches the alveolar space, it stays in contact with ALF without encountering alveolar resident cells for an extended period of time. The use of 15 min vs. 12 h exposure has been clarified in the Methods section (lines 461-467).
9. Figure 2: The footnote describes “n=2 biological replicates, from two independent ALF donor”. If it is correct, this study does not make good sense, though the findings are interesting.
Response: To clarify this statement, since we exposed M.tb to individual ALFs, we consider each ALF to be an independent biological replicate. In order to improve the robustness of the study and the conclusions, we have increased the number of ALFs tested to n=6 for the targeted qPCR assay (Results section 2.1), and the manuscript has been modified accordingly (lines 124-180). Since we are seeing consistent results for M.tb exposed to A-ALF vs. E-ALF (Results section 2.2) using n=3 ALF samples, we believe it is not necessary to increase the ‘n number’ in this set of experiments. We had several publications with n=3 (biological replicates) per group demonstrating significant differences.
10. In many elderly TB patients, reactivation of M. tuberculosis from latent state is believed to be the major process. In this context, what will be the meaning of weakened ALF in the reactivation process?
Response: Although here we are only studying the early M.tb-ALF interactions during primary M.tb infection, we can speculate that a dysfunctional ALF in the elderly during reactivation will be detrimental, where less ALF-driven alterations of the cell envelope will lead to exacerbated M.tb replication in the lungs and development of active TB. We have not included this in the discussion to avoid confusing the IJMS readers about the goal of this study, which is only focused on early M.tb-ALF interactions. Instead, we have indicated in the discussion that future studies will focus on the role of ALF during latency and TB reactivation (lines 358-359).
We hope that the reviewer find the changes to the manuscript appropriate and sufficient for publication.
Reviewer 2 Report
Dear authors,
Thank you for submit your valuable manuscript.
After careful reading, the overall manuscript on "Host- and age-dependent transcriptional changes in Mycobacterium tuberculosis cell envelope biosynthesis genes after exposure to human alveolar lining fluid" is comprehensive and very interesting in the contents and infographic figures. However, there are few comments (all are in yellow highlighted colors in the attached file) to be considered for improving this present manuscript. Please revise throughout the text to enhance the scientific values.
Sincerely,

Author Response
We thank the reviewer for the valuable comments to improve this manuscript. See below the changes made to the manuscript based on the reviewer’s suggestions:
- We have cited the references accordingly throughout the manuscript as suggested by the reviewer, adding the corresponding reference for each statement in a sentence when appropriate.
- In the methods, we have changed l > L (e.g. ml > mL) for consistency, as well as added company, city, and country information for all the reagents used.
- We have fixed the italics and lowercase characters in the References section.
- Text from lines 132-140 has been divided into two sentences. Since references 31 and 32 describe several of the genes and oligos used in the list of genes, we left them at the end of the sentence.
We hope that the reviewer find the changes to the manuscript appropriate and sufficient for publication.
This manuscript is a resubmission of an earlier submission. The following is a list of the peer review reports and author responses from that submission.
Round 1
Reviewer 1 Report
The submitted manuscript presents the results of gene transcription analysis of M.tuberculosis exposed to alveolar lining fluid from healthy subjects and from elder people. The study proves the changes in the bacteria metabolism which are induced by the treatment and lead to recompensation mechanisms.
The undertaken study has solid basis justified by previous studies. The working hypothesis is correct and it is verified within this manuscript. Experiments are correct. The presentation of results is detailed, supported by tables and figures. Discussion is current and extensive.
I have no critical comments.
Reviewer 2 Report
General comment:
The authors evaluated the targeted gene transcriptions of Mycobacterium tuberculosis affected by alveolar lining fluids of two generations, (young and old, relatively). The idea is quite good and the results are also interesting. However, the major limitation is the limited number of samples tested in this study. How do the authors generalise the results from two specimens each? How do the authors adjust the individual variation? In addition, the age is not an only factor to predict individual homeostasis. What biochemical factor(s) actually affected the results? The study requires more specimens and biochemical analysis for better understanding.
Specific comments:
1. What does the cell envelope mean in this study exactly? Does the concept include capsule or only so called mycomembrane?
2. The authors describe that ALFs were collected from BALF. However, BALF may contain not only alveolar lining fluid but also epithelial lining fluid. How did the authors prevent the contamination of fluid other than ALF? Did the authors confirm the biochemical components of the BALF specimen?
3. Are the mycomembrane structures of cultured M. tuberculosis and those in generated bioaerosol from active tuberculosis patient same? Is the use of cultured M. tuberculosis in this study relevant?
4. Figure 1 and Table 1 should move to materials and methods section.
5. It is not clear from this manuscript whether the authors mixed all ALF specimens and made up pooled stock or used individual ALF. It is surely important to make this point clear. If latter, how did the authors choose the ALFs use in this study?
6. There is no demographic or personal data of individuals from whom the ALF samples were collected. Did the authors perform IGRA to all participants; no IGRA positives?
7. Why did the authors perform targeted gene expression analysis? RNAseq will provide more deep insights about the influence of ALF to mycobacterial components, as the authors described in discussion section.
8. 15 min exposure seems reasonable, but 12 h seems too long. No cellar immunity response happens within 12 h in human alveolar region?
9. Figure 2: The footnote describes “n=2 biological replicates, from two independent ALF donor”. If it is correct, this study does not make good sense, though the findings are interesting.
10. In many elderly TB patients, reactivation of M. tuberculosis from latent state is believed to be the major process. In this context, what will be the meaning of weakened ALF in the reactivation process?
Reviewer 3 Report
Manuscript by Anna Allué-Guardia demonstrated that M.tb strains GFP-Erdman and H37Rv were exposed to ALFs and this exposure indicated that transcriptional changes are permanent since the changes sustained when ALF was removed.
Authors claimed that M.tb comes in close contact with the lung mucosa before and after its encounter with host alveolar compartment cells and that is the case of true infection – very interesting. Overall Anna Allué-Guardia concluded M.tb responds and adapts to the lung alveolar environment upon contact, and that the host ALF might play an important role in determining infection outcome.
The bacilli could represent LTBI in the lungs and this mimics the dormant bacteria. In the light of the above assumptions, authors are advised to do the following experiments-
- Grow both TB strains and expose to hypoxia following Wayne model and do ALF exposure to Dormant bacteria
- Grow both TB strains and expose them to Ascorbic acid (follow methods developed by Taneja et al PLOS ONE) and do ALF exposure and compare transcriptional changes
- The authors must also show the PDIM status in both strains Post-ALF exposure. The authors can run a TLC or similar chromatography experiment to show the PDIM status.